# Inclusion of Nature-Based Solution in the Evaluation of Slope Stability in Large Areas

Lukáš Zedek [1,*,†,‡], Jan Šembera [1,‡] and Jan Kurka [2,‡]

1   Institute of Novel Technologies and Applied Informatics, Faculty of Mechatronics, Informatics and Interedisciplinary Studies, Technical University of Liberec, Studentská 1402/2, 46117 Liberec, Czech Republic
2   AZ Consult, Spol. s r.o., Klíšská 12, 40001 Ústí nad Labem, Czech Republic
*   Correspondence: lukas.zedek@tul.cz
†   Current address: Institute of Novel Technologies and Applied Informatics, Faculty of Mechatronics, Informatics and Interedisciplinary Studies, Technical University of Liberec, Heydukova 1181/2, House C2, Office C02012, 46117 Liberec, Czech Republic.
‡   These authors contributed equally to this work.

**Abstract:** In areas affected by mining, which are undergoing reclamation, their geotechnical characteristics need to be monitored and the level of landslide risk should be assessed. This risk should preferably be reduced by nature-based solutions. This paper presents a KurZeS slope stability assessment technique based on areal data. This method is suitable for large areas. In addition, a procedure is presented for how to incorporate a prediction of the impact of nature-based solutions into this method, using the example of vegetation root reinforcement. The paper verifies the KurZeS method by comparing its results with the results of stability calculations by GEO5 software (version 5.2023.52.0) and validates the method by comparing its results with a map of closed areas in the area of the former open-cast mine Lohsa II in Lusatia, Germany. The original feature of the KurZeS method is the use of a pre-computed database. It allows the use of an original geometrical and geotechnical concept, where slope stability at each Test Point is evaluated not just along the fall line but also along different directions. This concept takes into account more slopes and assigns the Test Point the lowest safety factor in its vicinity. This could be important, especially in soil dumps with rugged terrain.

**Keywords:** slope failure risk prediction; slope stabilization; nature-based solutions; revegetation; root reinforcement





## 1. Introduction

Large-scale landslides can cause injuries, casualties, property damage and social as well as ecological problems [1]. Therefore, there is a strong need to prevent these events. This can be achieved and supported by an appropriate landslide susceptibility assessment, realized through a correctly selected evaluation method or software. The evaluation methods were intensively developed in the past decades. Nowadays, as mentioned in [2] and similarly in [3], they are usually divided into several classes: (1) qualitative methods that are based on expert experience, (2) semi-quantitative methods that are supplemented with scores/weights assigned to particularly assessed landslide susceptibility triggering aspects and (3) quantitative methods that can be further divided into geotechnical methods (physically based slope stability evaluation) [4–8], statistical analysis [2,9], neural network methods [10,11] or neuro-fuzzy logic methods [12,13].

In this paper, a physically based geotechnical model (a quantitative method) is considered. It is specifically a two- or three-layered geotechnical model with the use of the Bishop [14] (circular slip surface) and Sarma [15] methods (polygonal slip surface). In this paper, the method is referred to as KurZeS. Whereas many methods and software tools based on similar ideas use an assumption of infinite slope, e.g., [16–22], KurZeS does not, similarly to methods implemented in GRASS GIS [23–25], but it differs significantly in its

geometrical concept described in Section 2 and also in its technical implementation using interpolation based on pre-computed data. The method provides an alternative procedure for slope stability assessment in large areas and an original implementation allowing calculations to be performed on desktop computers with limited computing power.

Shallow landslides, in particular, that can be assessed as described in [16,18–20,22,26–29] can be partially avoided through nature-based solutions (NBS). Root reinforcement [30,31] is the particular effect that stabilizes the slope in a natural way. Root reinforcement has not been investigated since at least 70 years ago. It is usually simulated either through additional cohesion of the rooted top soil layer [32] or through the Finite Element Method [33].

In KurZeS, the root reinforcement effect is simulated using the additional root cohesion concept. Evaluation of the additional root cohesion was described in [32,34–37]. KurZeS uses approximate values derived from values mentioned in [34,35,37].

For the visualization of the inputs and outputs of the model, we used the freely available geographic information system QGIS [38].

This article aims to analyze the influence of selected, nature-based solutions on the determination of the approximate slope stability on a large area. As typical examples, one can take areas intended for reclamation after open-pit coal mining. In the case of large areas, the prevailing reclamation method is afforestation or covering with other vegetation. Tree species recommended for reclamation of brown coal dumps are birch (*Betula*), larch (*Larix*), red oak (*Quercus rubra*) and pine (*Pinus*) [39]. In such a case, the main stabilization effect comes from the rooting of the surface layer.

The analysis is carried out in stages as follows:

1.  Evaluation of the top layer shear parameter increase due to the effect of rooting. This stage was based on the available literature data.
2.  The application of increased parameters in the large area surface stability model and realization of calculations using the original method created by the authors of the article.

The article is structured in five numbered sections. The first, the Introduction section, includes the background and context of the presented method. The second, the Materials and Methods section, is devoted to the description of the KurZeS method itself, its features and problems, the inclusion of nature-based solutions, the description of the testing problem and the description of the verification and validation approach. The third, the Results section, includes the results of the testing problem and results of the verification computations of specific details, and their comparison. The fourth, the Discussion section, includes specific observations concerning the KurZeS method and the inclusion of the NBS in it. The last, the Conclusions section, summarizes the main results and observations of the article.

## 2. Materials and Methods

### 2.1. Study Area

For this study, the area of the soil dump of the former open-cast mine Lohsa II in Lusatia, Germany, see Figure 1, was chosen. The map presented below was obtained from [40]. The area is interesting because several landslides have appeared there in recent times. Specifically, the slopes of the mining waste dump Scheibe at the northwest border of the closed coal mine Lohsa II moved in the spring of 2019. Even later (in the spring of 2021), landslides appeared on the slopes near the lake Knappensee.

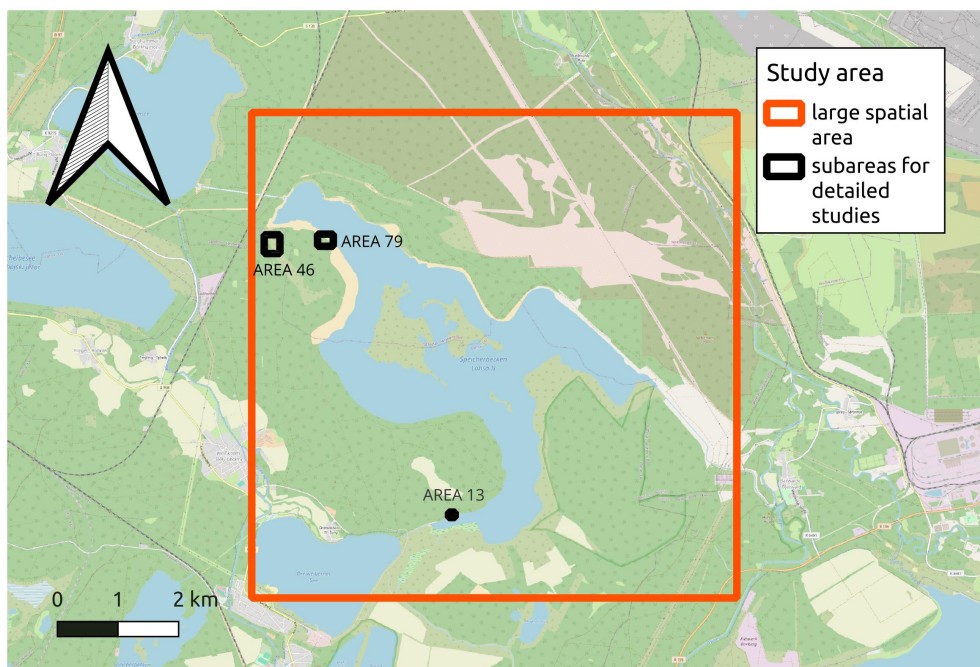

**Figure 1.** Study area selected for slope stability assessment (red) and three subareas for verification of results.

*2.2. Geometrical Concept*

The used data describing the digital elevation model (DEM) of the terrain within the study area are organized in triplets of coordinates [X; Y; Z] with a regular step of 10 m on both the X and Y axes. The safety factor is evaluated in each point of the 10 × 10 m raster (this point is referred to as the TP—the Test Point). The safety factor corresponding to the TP is derived as the minimal safety factor of all relevant slopes in the TP's neighborhood. The slopes relevant to the TP are defined by pairs of boundary points ($BP_1$ and $BP_2$) satisfying the following conditions:

- The boundary points are different points located at the points of the 10 × 10 m raster not further than 100 m from the TP.
- The TP may coincide with one of the boundary points.
- The total length of the (possibly angled) line $BP_1 - TP - BP_2$ must be at least 20 m and at most 100 m.
- The convex obtuse angle of line segments $BP_1 - TP$ and the $TP - BP_2$ shall be greater than or equal to 120°.

The safety factor of the TP is then evaluated as the minimum of the safety factors of all slopes between $BP_1$ and $BP_2$ satisfying the above-specified conditions. The algorithm KurZeS includes an optimization trick based on the obvious equivalence of the slope $BP_1 - BP_2$ and $BP_2 - BP_1$ reducing the number of safety factor estimations to a half (see Figure 2).

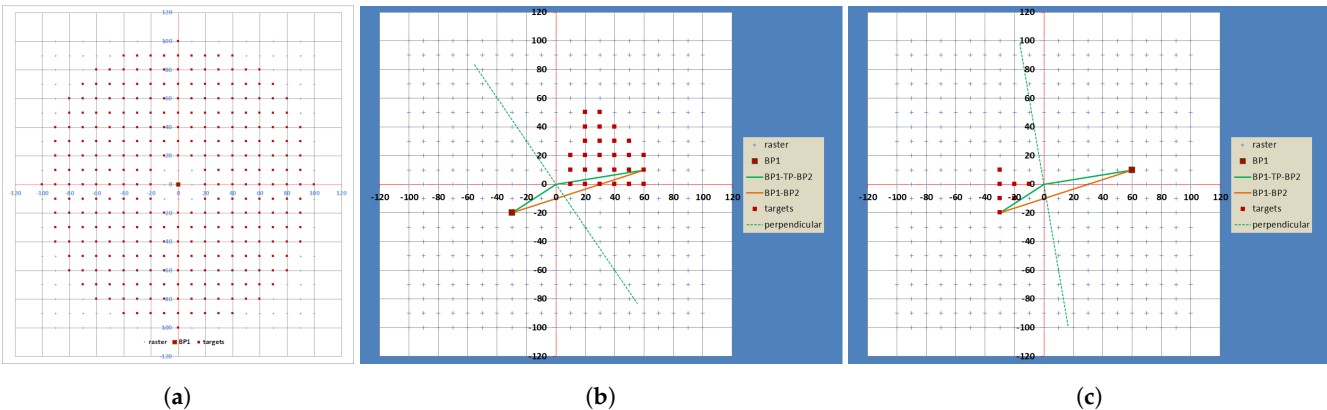

(a)            (b)            (c)

**Figure 2.** (**a**) The set of endpoints for $BP_1 \equiv TP$; (**b**,**c**) example of repetition of found pair $BP_1$ and $BP_2$ in case $BP_1 \neq TB \neq PB_2$.

### 2.3. Geotechnical Concept

The safety factor of each slope $BP_1 - BP_2$ is evaluated with interpolation from a pre-calculated database based on the following geotechnical model constructed in GEO5 [41] (version 5.2023.52.0).

#### 2.3.1. Model

The basic model is constructed as two-layered, with a constant gradient of the terrain in variants in the range from 1 in 2 to 1 in 10. The interface between the soil layers runs parallel to the terrain at a depth of 5 m. The parameters of the top layer are exactly those of the soil it contains. The underlying layer is supposed to consist of the same type of soil but firmer, which is modeled by higher stability parameters compared to the top layer. This increase reflects the usual improvement with depth (see Figure 3a and Table 1).

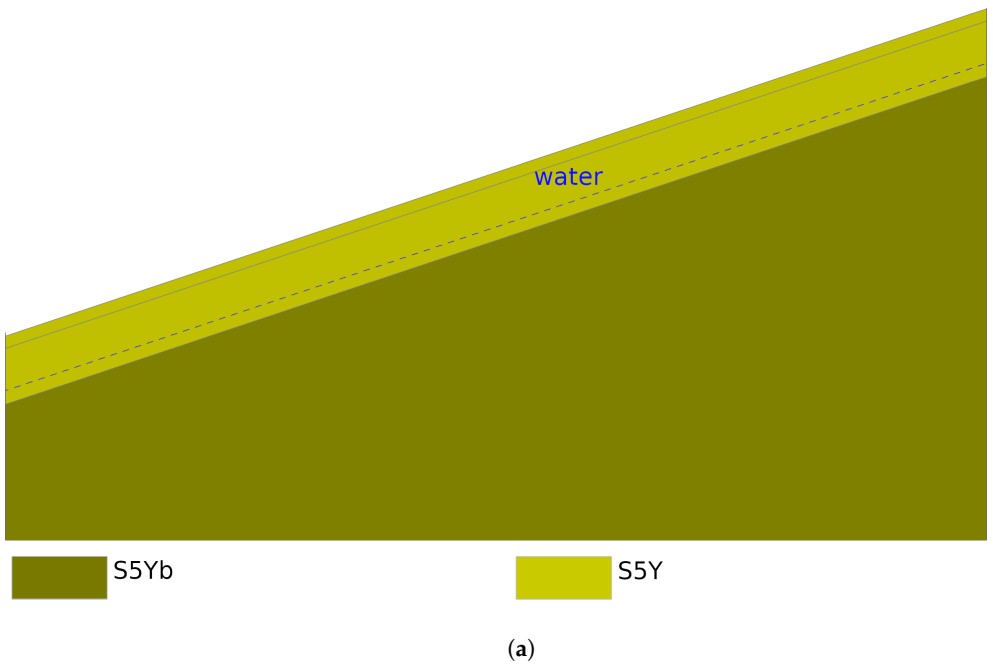

(**a**)

**Figure 3.** *Cont.*

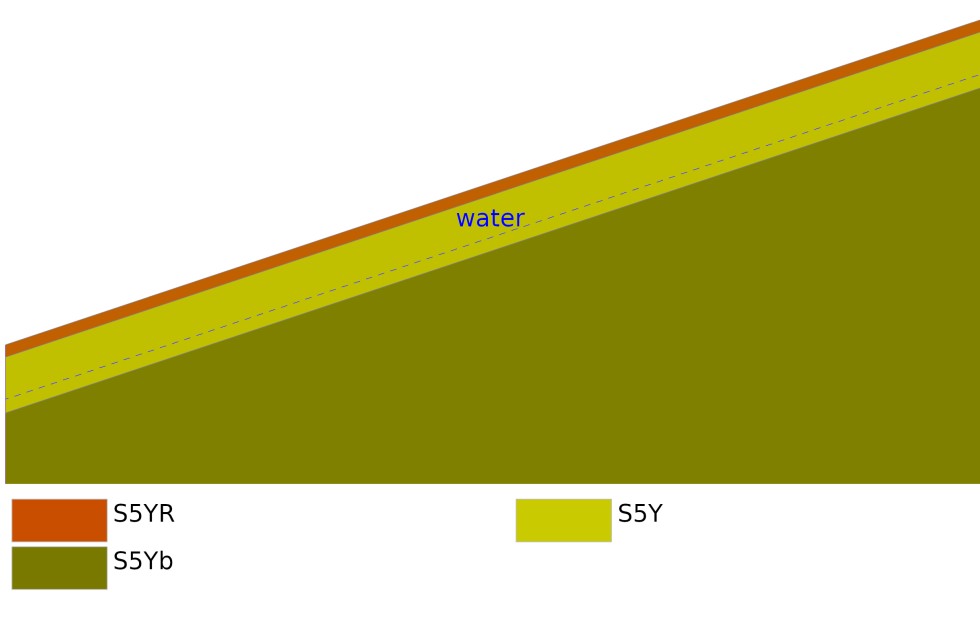

(**b**)

**Figure 3.** (**a**) Two-layer model; (**b**) three-layer model.

A more complex model is used for the case of root reinforcement. Here, the third layer of a constant thickness of 0.9 m is added to the surface (see Figure 3b and Table 1). The used soil parameters in it are also higher than in the original soil, this time due to root reinforcement. The specific values of increase are derived from the literature [34,35,37].

The groundwater level is considered parallel to the surface in variants from a depth of −20 to 0 m.

According to the method used, circular (Bishop method [14]) and polygonal (Sarma method [15]) slip surfaces with a fixed length from 20 to 100 m are considered.

2.3.2. Resulting Database

The result of an individual stability calculation in the given model is the search for the least favorable slip surface (either circular or polygonal, using the optimization procedure) and the determination of the safety factor $F_s$ by the limit equilibrium method (LEM). $F_s$ values were calculated for all options including 8 slope inclinations, 6 slope lengths, 7 groundwater levels and 2 cases with and without a rooted topsoil layer. For each selected soil type, the results database consists of 672 $F_s$ values.

The safety factor of a specific slope $BP_1 − BP_2$ is derived from this database so that the value of the corresponding slope inclination, slope length and groundwater level of the slope $BP_1 − BP_2$ is specified and the $F_s$ value is interpolated from neighboring values tabulated in the database.

It is typical for the particular two- or three-layer model with parallel construction described above that the type and shape of the slip surface changes as the least favorable slip surface lengthens. Short slip surfaces with the lowest $F_s$ are typically circular in shape (B—Bishop) confined in the uppermost layer, or in the central and rooted layer, not extending into the lowest firmer layer with an interface 5 m below the surface (Figure 4a). The intermediate slip surfaces are most commonly polygonal (S—Sarma) following the top of the interface at 5 m depth. In terms of type, they most closely resemble the infinite slope model (Figure 4b). In some cases, usually with the groundwater level between depths of 3 and 6 m below ground level, long slip surfaces of circular shape cross the aforementioned interface at a depth of 5 m below the surface and extend deep into the firmer layer (BD—Bishop, deep)—Figure 4c.

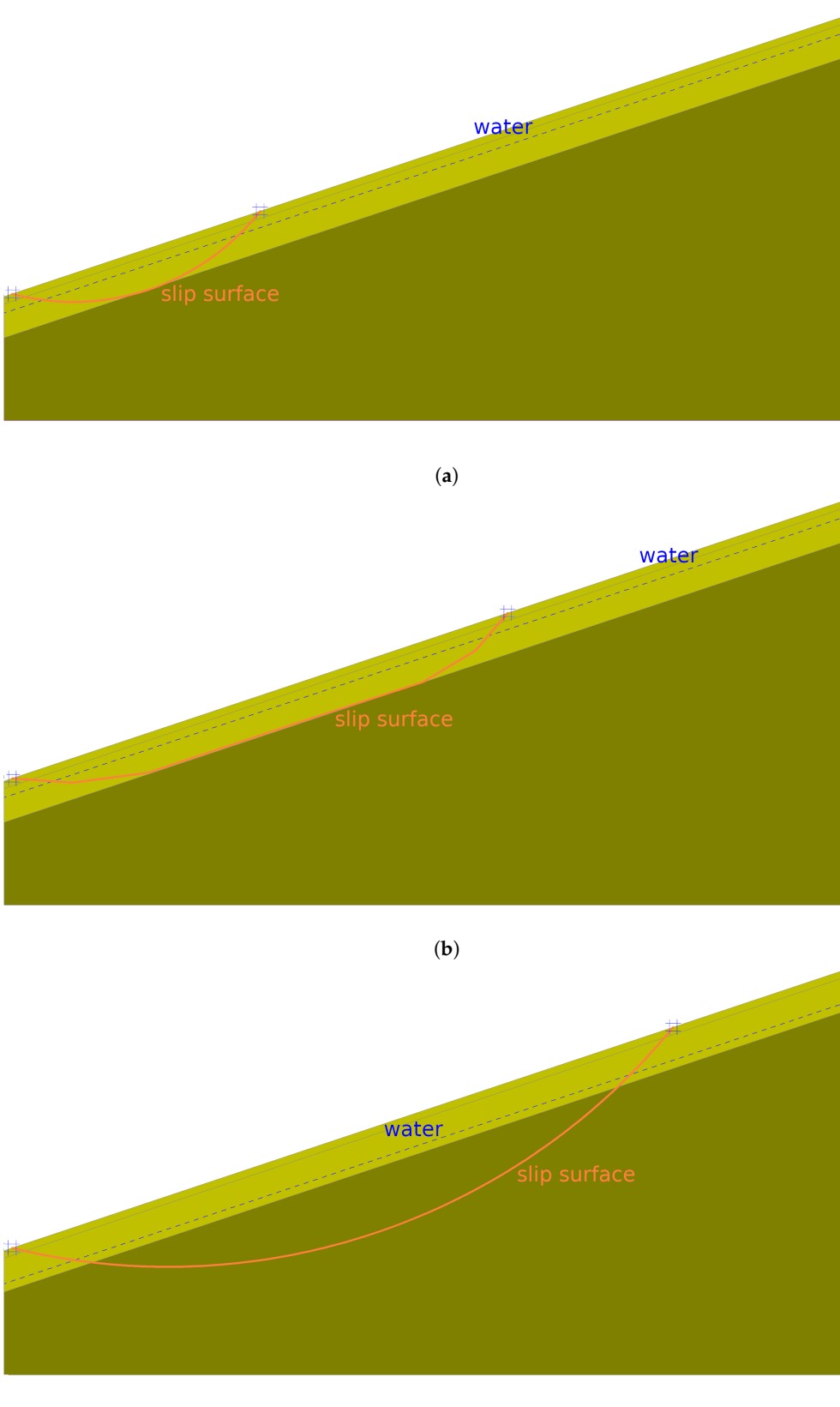

**Figure 4.** (**a**) Short circular slip surface; (**b**) medium-length polygonal slip surface; (**c**) long deep circular slip surface.

It is not clear in advance which of the slip surfaces will be the least favorable. The result is mainly influenced by the geometry of the model, soil parameters and the position of the water table or surface water level. It should be emphasized repeatedly that the results are based on the chosen model; for a differently designed model, the composition of the least favorable slip surfaces could be different.

### 2.4. Quantification of the Effect of Rooting

Figure 5 shows an example from the resulting database of $F_s$ for one selected soil type denoted as S5Y (sandy dump). The left half of the table applies to the model without a rooted topsoil layer, and the other to the model with a rooted layer. In the header of the table, the parameters used for effective strength (effective angle of internal friction (°) and cohesion (kPa)) and bulk weight (kN/m$^3$) are typed. The first column shows the depth of the water table, the second column shows the slope inclination and the first row under the heading shows the fixed length of the slip surface. The types of slip surfaces in the sense of the text above are distinguished by color for cells with the resulting safety factors $F_s$ (short circular slip surface in blue, medium-length polygonal slip surface in green and deep circular slip surface in red).

| S5Y | | $\phi_{eff}$ | $c_{eff}$ | $\gamma$ | $\gamma_{sat}$ | | | S5Y | | $\phi_{eff}$ | $c_{eff}$ | $\gamma$ | $\gamma_{sat}$ | | |
|---|---|---|---|---|---|---|---|---|---|---|---|---|---|---|---|
| | 1 | 25 | 2 | 19 | 19.5 | | | | 1 | 25 | 2 | 19 | 19.5 | | |
| | 1plus | 29 | 4 | 20 | 20.5 | | | | 1plus | 29 | 4 | 20 | 20.5 | | |
| | | 25 | 18 | 19 | 19.5 | | | root | | 25 | 18 | 19 | 19.5 | | |
| gndwater | 1 in ? | 20 | 30 | 40 | 60 | 80 | 100 | gndwater | 1 in ? | 20 | 30 | 40 | 60 | 80 | 100 |
| -20 | 3 | 1.75 | 1.66 | 1.62 | 1.56 | 1.54 | 1.52 | -20 | 3 | 2.01 | 1.82 | 1.72 | 1.62 | 1.58 | 1.55 |
| -6 | 3 | 1.75 | 1.67 | 1.62 | 1.56 | 1.53 | 1.49 | -6 | 3 | 2.01 | 1.82 | 1.72 | 1.62 | 1.58 | 1.50 |
| -4 | 3 | 1.75 | 1.66 | 1.57 | 1.49 | 1.43 | 1.35 | -4 | 3 | 2.02 | 1.77 | 1.66 | 1.55 | 1.44 | 1.36 |
| -3 | 3 | 1.75 | 1.53 | 1.44 | 1.35 | 1.29 | 1.27 | -3 | 3 | 1.93 | 1.63 | 1.53 | 1.40 | 1.33 | 1.28 |
| -2 | 3 | 1.58 | 1.36 | 1.28 | 1.18 | 1.13 | 1.12 | -2 | 3 | 1.71 | 1.46 | 1.36 | 1.23 | 1.17 | 1.14 |
| -1 | 3 | 1.31 | 1.16 | 1.08 | 1.01 | 0.99 | 0.96 | -1 | 3 | 1.46 | 1.25 | 1.17 | 1.06 | 1.02 | 0.98 |
| 0 | 3 | 0.97 | 0.90 | 0.86 | 0.82 | 0.80 | 0.79 | 0 | 3 | 1.18 | 1.02 | 0.95 | 0.89 | 0.84 | 0.81 |
| -20 | 4 | 2.33 | 2.22 | 2.15 | 2.08 | 2.04 | 2.02 | -20 | 4 | 2.70 | 2.43 | 2.30 | 2.17 | 2.10 | 2.08 |
| -6 | 4 | 2.33 | 2.22 | 2.15 | 2.08 | 2.04 | 2.00 | -6 | 4 | 2.70 | 2.42 | 2.30 | 2.17 | 2.10 | 2.01 |
| -4 | 4 | 2.33 | 2.22 | 2.09 | 1.96 | 1.90 | 1.80 | -4 | 4 | 2.70 | 2.36 | 2.21 | 2.02 | 1.93 | 1.81 |
| -3 | 4 | 2.33 | 2.05 | 1.92 | 1.78 | 1.73 | 1.69 | -3 | 4 | 2.57 | 2.17 | 2.03 | 1.84 | 1.78 | 1.71 |
| -2 | 4 | 2.11 | 1.82 | 1.70 | 1.55 | 1.51 | 1.50 | -2 | 4 | 2.30 | 1.94 | 1.81 | 1.61 | 1.56 | 1.54 |
| -1 | 4 | 1.76 | 1.54 | 1.44 | 1.34 | 1.30 | 1.27 | -1 | 4 | 1.96 | 1.66 | 1.55 | 1.41 | 1.35 | 1.31 |
| 0 | 4 | 1.29 | 1.19 | 1.14 | 1.09 | 1.06 | 1.05 | 0 | 4 | 1.56 | 1.34 | 1.26 | 1.16 | 1.16 | 1.09 |
| -20 | 5 | 2.90 | 2.76 | 2.69 | 2.60 | 2.55 | 2.52 | -20 | 5 | 3.35 | 3.03 | 2.87 | 2.69 | 2.62 | 2.57 |
| -6 | 5 | 2.90 | 2.76 | 2.69 | 2.60 | 2.55 | 2.50 | -6 | 5 | 3.35 | 3.04 | 2.87 | 2.69 | 2.62 | 2.51 |
| -4 | 5 | 2.90 | 2.76 | 2.62 | 2.46 | 2.39 | 2.25 | -4 | 5 | 3.35 | 2.95 | 2.76 | 2.54 | 2.41 | 2.27 |
| -3 | 5 | 2.90 | 2.57 | 2.40 | 2.23 | 2.14 | 2.11 | -3 | 5 | 3.20 | 2.73 | 2.54 | 2.32 | 2.20 | 2.13 |
| -2 | 5 | 2.63 | 2.28 | 2.12 | 1.98 | 1.89 | 1.86 | -2 | 5 | 2.86 | 2.42 | 2.24 | 2.06 | 1.94 | 1.90 |
| -1 | 5 | 2.18 | 1.93 | 1.80 | 1.68 | 1.62 | 1.60 | -1 | 5 | 2.42 | 2.08 | 1.92 | 1.77 | 1.67 | 1.65 |
| 0 | 5 | 1.60 | 1.49 | 1.42 | 1.36 | 1.32 | 1.30 | 0 | 5 | 1.92 | 1.68 | 1.57 | 1.45 | 1.39 | 1.35 |
| -20 | 6 | 3.48 | 3.32 | 3.22 | 3.12 | 3.06 | 3.02 | -20 | 6 | 4.05 | 3.63 | 3.42 | 3.23 | 3.14 | 3.09 |

**Figure 5.** Example from the resulting database of $F_s$ for S5Y soil (sandy dump).

### 2.5. Data Sets

All data sets were obtained from freely available sources and they were transformed into the form of georeferenced rasters, where the centers of gravity of particular raster cells are the nodes of the network described in Section 2.2 dedicated to the geometrical concept of the model.

### 2.5.1. Digital Elevation Model

The main data source for performed simulations was the digital elevation model (DEM) DGM1 (German DEM [42]). This DEM (multiple shape files) was rasterized with the use of QGIS [38]. The selected cell edge length was 10 m. This selection of the cell edge length brought a compromise between acceptable computation times (huge number of elements/nodes) and sufficient accuracy of the slope description, see Figure 6.

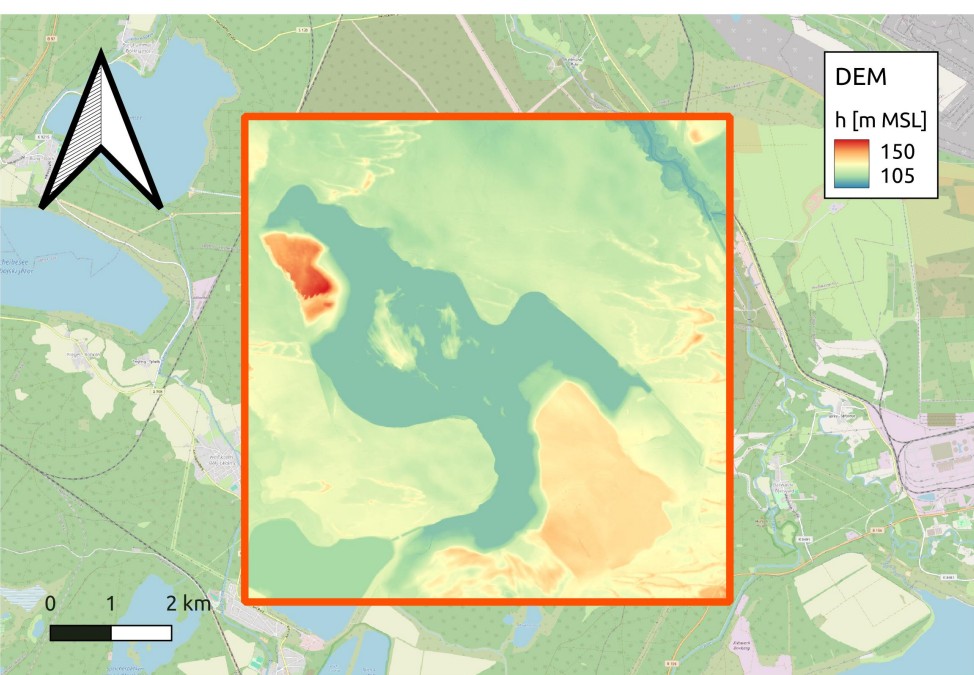

**Figure 6.** Digital elevation model rasterized with cell edge length 10.0 m.

### 2.5.2. Ground Water Table

Because of missing data, the groundwater table was derived from the DEM. For this purpose, raster calculator "gdal_calc", which is a part of GDAL-extension [43] for QGIS, was used together with Equation (1), where W means water elevation, A represents a surface elevation in DEM-raster, 115 stands for the selected lake water level, 0.75 defines the "speed" of water table increase and 20 and 0 are the maximal and minimal groundwater table depths, respectively.

$$W = \begin{cases} 115 & \text{if } A \le 115 \\ A - 20 & \text{if } (A - 115) \cdot (1 - 0.75) > 20 \\ 115 + (A - 115) \cdot 0.75 & \text{otherwise} \end{cases} \tag{1}$$

The resulting water table raster is shown in Figure 7.

### 2.5.3. Type of Soils

A pedological map was obtained from LfULG (Bodenkarte BK50 [44]). Shapefiles from the original map were merged and divided into two or four (non-rooted soil and rooted soil, respectively) different groups, and they were rasterized in the same spatial extent and with the same cell size (10 × 10 m) as the DEM, see Figure 8a. For the case of rooted soils, two further soil types appeared, see Figure 8b. The distribution of rooted and non-rooted soil was based on the difference between the DEM and the selected lake water level (115 m MSL). Rooted soils were considered for positive values (above water) and non-rooted soils for negative (below water).

The strength parameters of soils can be found in Table 1.

**Table 1.** The strength parameters of used soil types. $\varphi_{eff}$ denotes effective angle of internal friction and $c_{eff}$ stands for effective cohesion of soil. The thickness of the topsoil layer is 5 m and there is no rooted soil layer in case of no measures. In the case of 0.9 m rooting depth, the thickness of the rooted soil layer is 0.9 m and the thickness of topsoil layer is 4.1 m. Parameter values are based on expert judgment combined with information from the literature [34,35,37].

| | Basal Soil Layer | | Topsoil Layer | | Rooted Soil Layer | |
| --- | --- | --- | --- | --- | --- | --- |
| | $\varphi_{eff}$ | $c_{eff}$ | $\varphi_{eff}$ | $c_{eff}$ | $\varphi_{eff}$ | $c_{eff}$ |
| S45 * | 31° | 16 kPa | 27° | 8 kPa | 27° | 24 kPa |
| S5Y * | 29° | 4 kPa | 25° | 2 kPa | 25° | 18 kPa |

* The soil type notation comes from the Czech Technical Standard ČSN P 73 1005 (731005) Engineering geological survey [45]; the Y symbol denotes the relocated material in non-original placement.

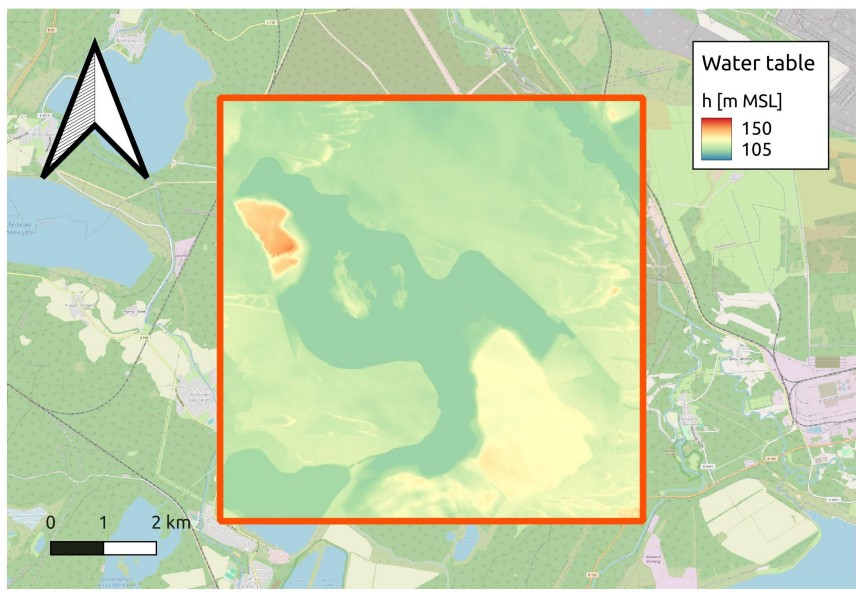

**Figure 7.** Linearly increasing water table.

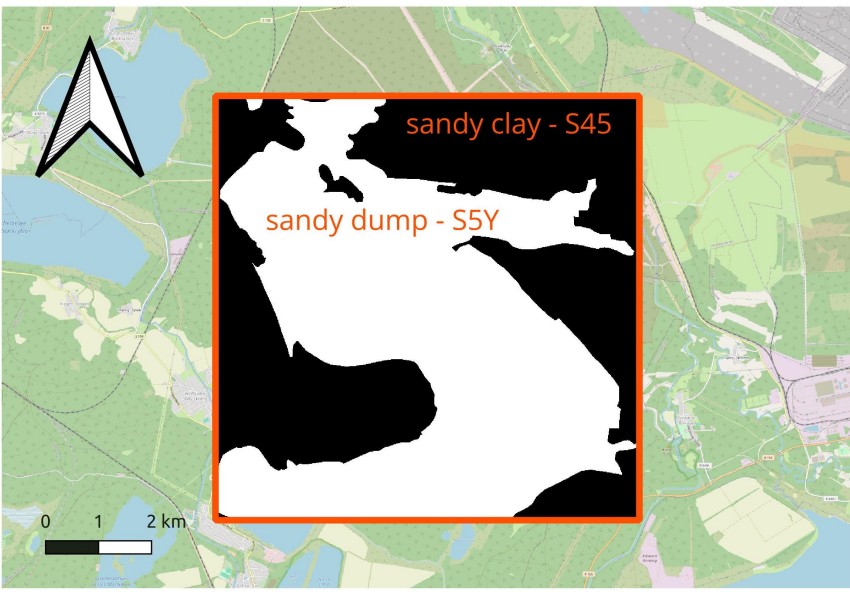

(**a**)

**Figure 8.** *Cont.*

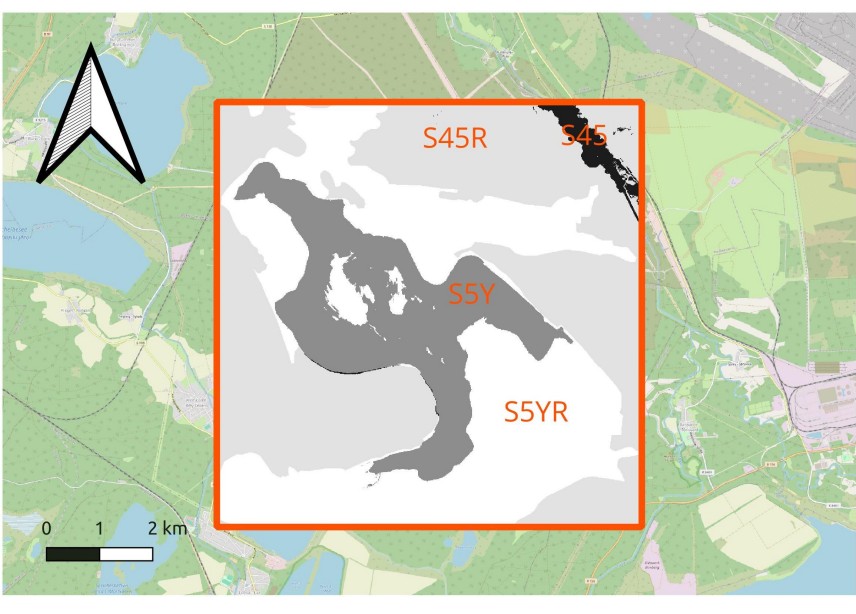

(**b**)

**Figure 8.** Simplified categorization of soil types. (**a**) Non-rooted soil types. (**b**) Soil types without (under the water) and with roots.

*2.6. Implementation*

Slope stability was derived from interpolation tables (resulting database, see Section 2.3.2) pre-computed in the GEO5 software [41] (version 5.2023.52.0). derived the safety factor from the slope length (in range 20 to 100 m), slope inclination (in range 1:10 to 1:2), soil type (2 types of rooted soil and 2 types without reinforcement) and depth of the water table (in range 0 to 20 m).

Two variants of interpolation were tested: linear interpolation and polynomial interpolation. As they had comparable results and linear interpolation was noticeably faster, the presented computations were performed using multidimensional linear interpolation with the use of the linearNDInterpolator implemented in SciPy [46].

The calculations work with a circular or polygonal slip surface of general depth. Such a slip surface crosses the root-reinforced layer only along the possible landslide scarp. Thus, its area is the only place where the improvement in soil properties is accounted for by this method. In this way, the application of root reinforcement differs significantly from commonly used infinite slope stability models, e.g., [47].

For the presented computations, a computer with the following parameters was used:

- Intel® Core™ i7-3770K CPU; 3.50 GHz × 4;
- RAM 15.5 GiB;
- Ubuntu 22.04.4 LTS;
- Python 3.8.16.

Computation for such a large-scale area was not possible at once because of extensive memory demands. The results were obtained by dividing the area into 36 overlapping subareas (see Figure 9), which effectively decreased the memory requirements. Eventually, the computation could be realized in four parallel threads, i.e., in four subareas at the same time. A 100 m overlapping of subareas was necessary because of the geometrical concept of the model, see Section 2.2. One turn (4 subareas) took 14 min.

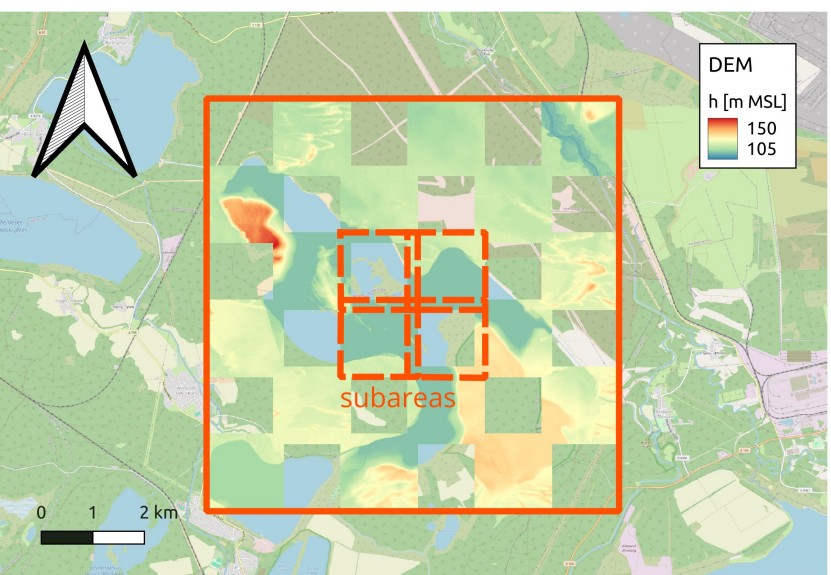

**Figure 9.** Division of the study area enabled parallel computation.

*2.7. Verification*

The mechanism for verification of the results can be found in Figure 10. Two safety factor computations were performed—one without root reinforcement and another with root reinforcement under consideration. Two hypotheses were tested as follows:

- Because all $F_s$ in the pre-calculated database are higher for root-reinforced soils than for the soils without vegetation, the safety factor should be higher or equal in each Test Point of prediction considering root reinforcement than in prediction without root reinforcement.
- The results of the prediction should be comparable with $F_s$ predicted by GEO5 at each slope in the area.

Several slopes in the area were taken, the surface and groundwater table cuts were taken from QGIS and included in GEO5 software and the following observations were made:

- Consideration of root reinforcement brought either the same or higher value of the predicted safety factor in every Test Point.
- For selected cuts, the predicted safety factor corresponded closely to the values obtained from GEO5.

For more details, see Section 3.

The two above-mentioned points verified the expected behaviour of the simulation code.

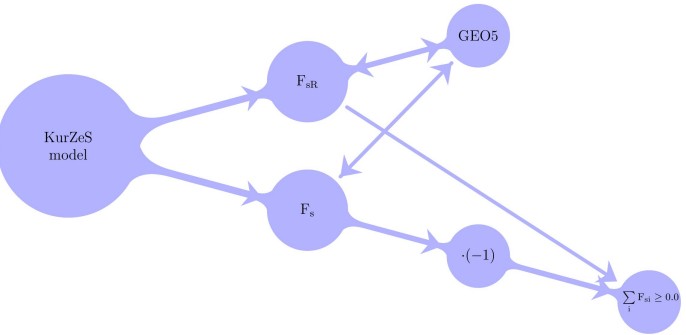

**Figure 10.** Verification diagram.

## 2.8. Validation

The approach for validation of the results is shown in Figure 11. The computed safety factor for all the cells in the study area with and without additional root cohesion under consideration was imported to QGIS [38] and compared with the shape file map of the area closed to the public because of safety reasons. The closed area correlates well with the slopes with a predicted $F_s$ very close to one, which validates the results with only independent accessible public data.

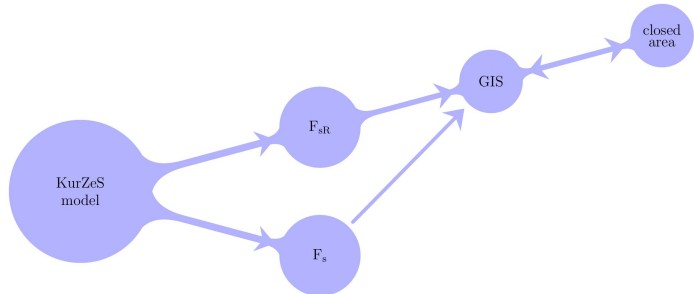

**Figure 11.** Validation diagram.

Another method of validation was the comparison of landslide risk maps with orthophoto maps of the study area and also with maps showing the slope gradient there. A noticeable correspondence was observed between these data, but due to the unclear, precise dating and spatial location of the landslides, these comparisons were not included in the study.

## 3. Results

### 3.1. Slope Stability Improvement Predicted by GEO5

To get a better idea about the expectable predicted safety factor improvement caused by additional root cohesion, selected data sets from the resulting database were visualized and analyzed. Safety factors of two selected soil types present in the study area, sandy clay (S45) and sandy dump (S5Y), each for one slope inclination, are depicted with and without vegetation cover (root reinforcement) in Figure 12a,b.

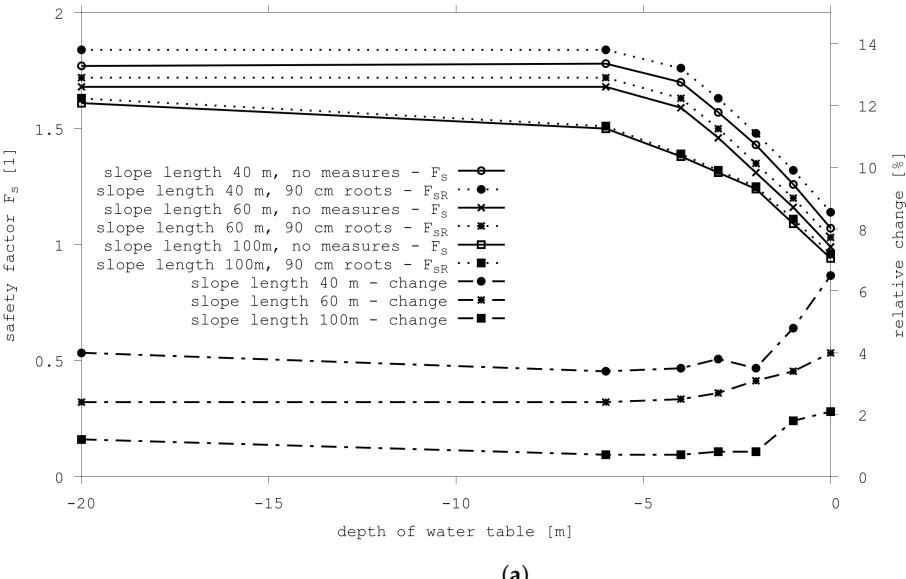

(**a**)

**Figure 12.** *Cont.*

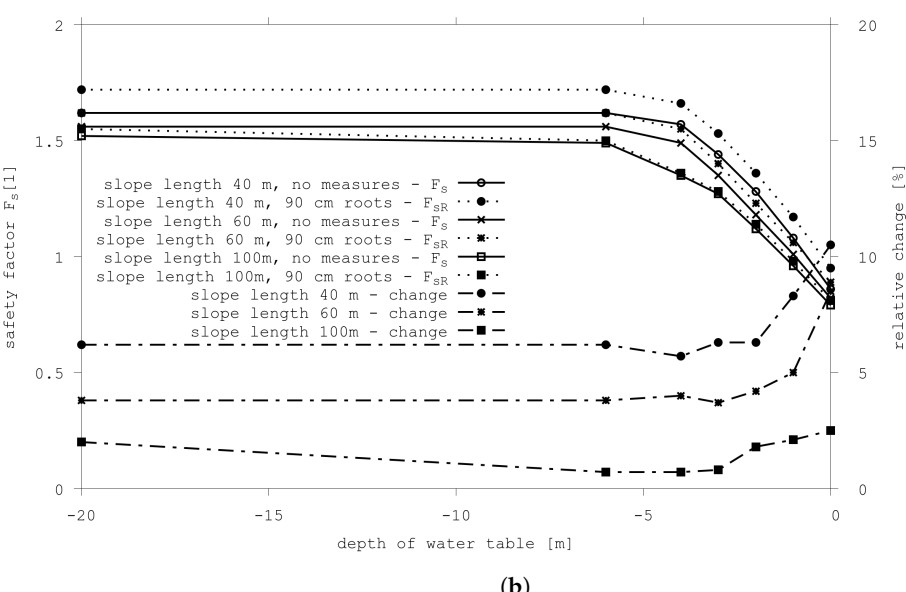

(**b**)

**Figure 12.** Comparison of predicted slope stability. (**a**) Sandy clay (S45), slope inclination 1 in 2.5 and three different slope lengths. (**b**) Sandy dump (S5Y), slope inclination 1 in 3 and three different slope lengths.

Both Figure 12a,b show the dependency of the absolute and relative improvement of the predicted safety factor on the depth of the water table as well as on the root reinforcement under consideration. Safety factor values (left-hand axis), as well as relative improvement (right-hand axis), decrease with growing slope length. The effect of root reinforcement is stronger in the case of sandy dump. However small the absolute improvement may seem, it can be crucial for the cases where driving and resisting forces are near to equilibrium (i.e., for $F_s$ around 1). The most significant relative improvement is predicted for slopes where groundwater reaches near the earth's surface, i.e., the least stable slopes.

### 3.2. Slope Stability Evaluation with KurZeS

As a first step for the testing of KurZeS, the predicted safety factor $F_s$ without the root reinforcement (additional cohesion) under consideration was evaluated, see Figure 13. The most landslide-threatened slopes appeared on the lake shore (e.g., "AREA 79" on the Figures 1) and on the steep slopes in the northwestern part of the waste dump Scheibe ("AREA 46" on the Figure 1).

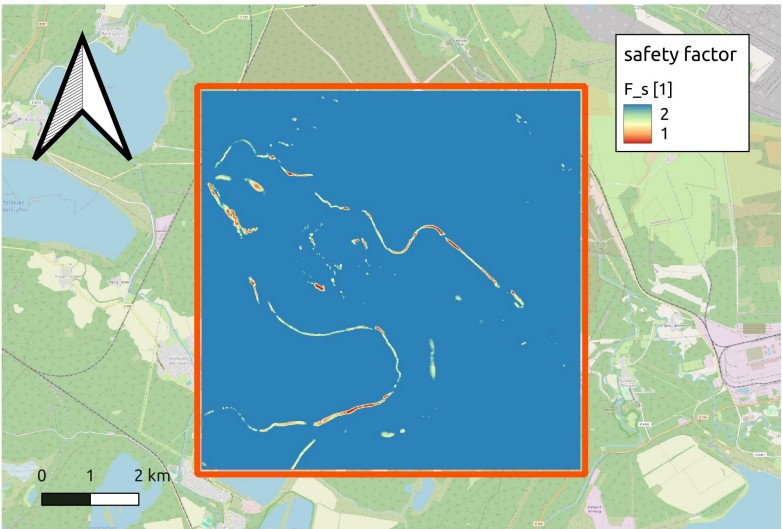

**Figure 13.** Slope stability evaluation without root reinforcement under consideration.

Figure 14 shows the comparison of the predicted safety factor of slopes for non-rooted soils with a shape file map of the area closed to the public. It shows good correspondence between both data sets and it shows the beneficial effect of the KurZeS approach for the identification of unstable areas suggested for further in situ investigation.

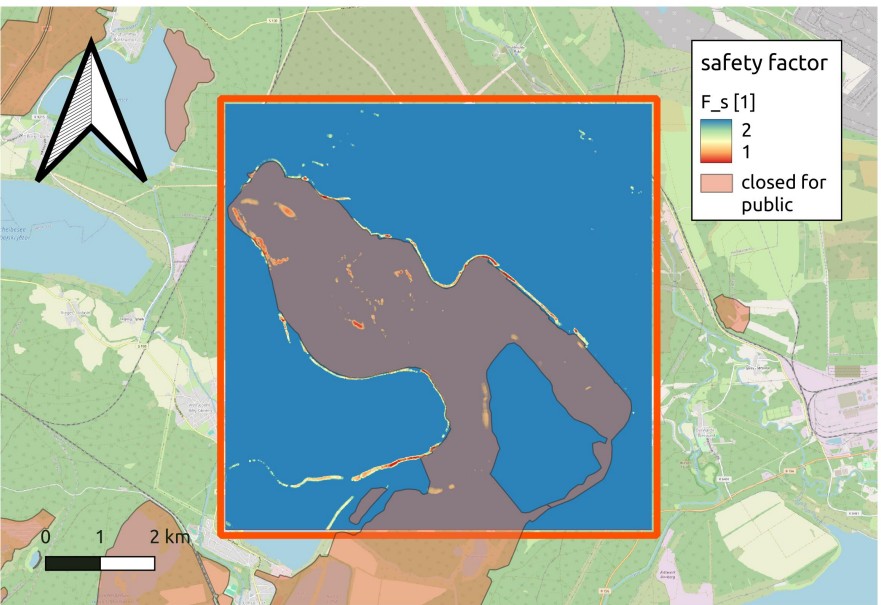

**Figure 14.** Comparison of results with shape file mapping area closed for public.

Another prediction was made with the addition of the root reinforcement under consideration. The resulting prediction of the safety factor is denoted as $F_{sR}$. The result can be seen in Figure 15. The values of $F_{sR}$ are generally higher (better) than the values of $F_s$ (without root reinforcement). It will be further analyzed in the following Section.

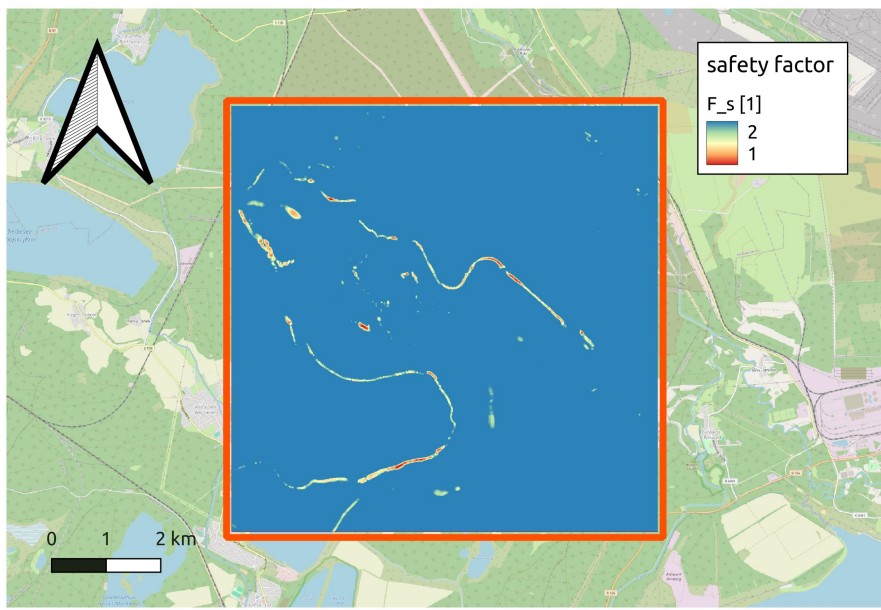

**Figure 15.** Slope stability evaluation with root reinforcement under consideration.

### 3.3. Improvement in Slope Stability Predicted by KurZeS Due to Root Reinforcement

From raster outputs of slope stability analyses, absolute and relative improvement in the predicted safety factor obtained with or without the roots under consideration was eval-

uated. The absolute improvement $F_{sR} - F_s$ reaches values in the range $\langle 0.0, 0.25 \rangle$. Relative improvement of the predicted safety factor $F_{sR}/F_s - 1$ depicted in Figure 16 reaches values in the range $\langle 0.0, 17.7\% \rangle$. Both the maximal values overreach the expectations constructed on the base from Section 3.1. The explanation coming from further analysis resides in the fact that the best improvement appears for extremely short slopes, which were not considered in Section 3.1.

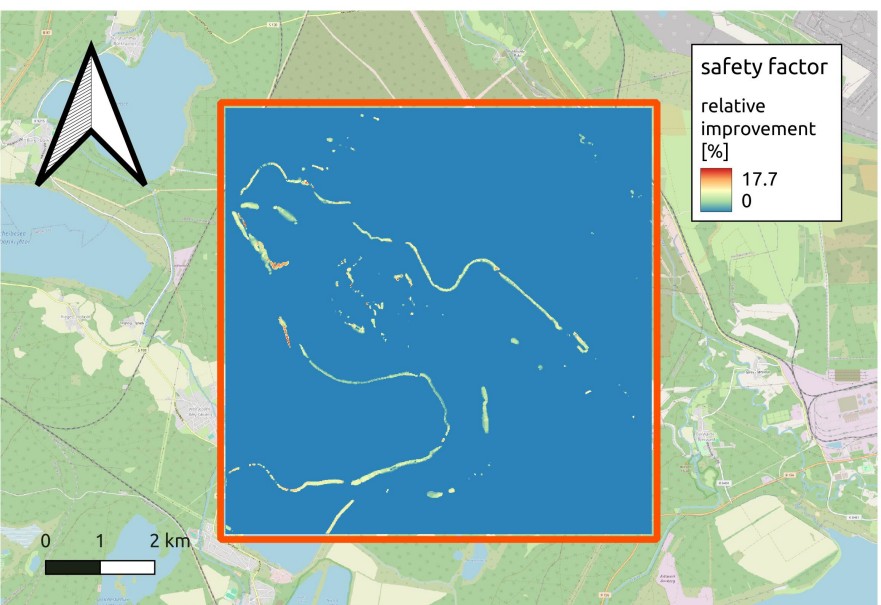

**Figure 16.** Relative improvement of slope stability due to root reinforcement.

### *3.4. Comparison of Results from KurZeS and GEO5*

To verify the results of slope stability evaluation by the KurZeS approach, several further studies of slope stability, in particular mesh cells, were performed. The KurZeS results were compared with those obtained from GEO5 for the particular slopes. In this section, we present two representative examples as well as data sets prepared for the following research.

For nine Test Points, the boundary points $BP_1$ and $BP_2$ realizing the minimal associated $F_s$ were identified. The connection $BP_1 - BP_2$ was prolonged in QGIS [38] and on such a cross-section, the model input data (DEM, depth of water table, soil type) were extracted from intersected raster cells. These cross-section data served as inputs to GEO5 software for cross-sectional safety factor evaluations. The Test Points were selected from three areas that were somehow interesting. The areas were named "AREA 13" (it included Test Points 1 to 3 and corresponding cross-sections 1 to 3), "AREA 46" (it included Test Points 4 to 6 and corresponding cross-sections 4 to 6) and "AREA 79" (which included Test Points 7 to 9 and corresponding cross-sections 7 to 9). Here, we present the results of cross-sections 5 and 8 that represent all nine cross-sections well.

#### 3.4.1. Cross-Section 5

In Figure 17, you can see the map from [40] of "AREA 46" that includes very steep slopes that are not at the lake shore with well-defined slope direction (almost planar slopes of a long straight valley). Test Point $TP_5$ that is not a part of the cross-section was selected. In Figure 17, in red, 13 sampling points ($SP\_i, j$) formed the cross-section model in GEO5.

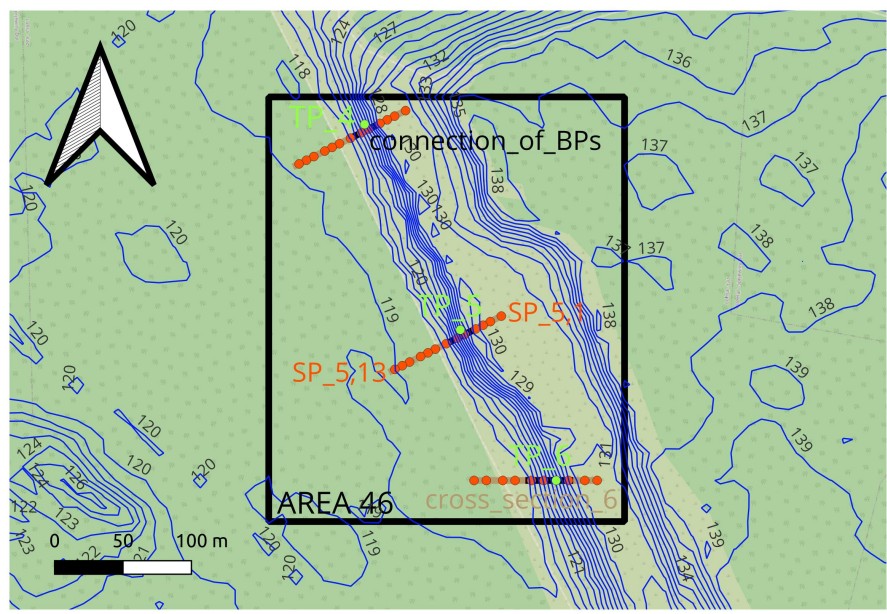

**Figure 17.** Map of "AREA 46".

For the non-rooted soils, KurZeS evaluated the safety factor $F_s(TP_5) = 1.10$. Results for the whole "AREA 46" are shown in Figure 18.

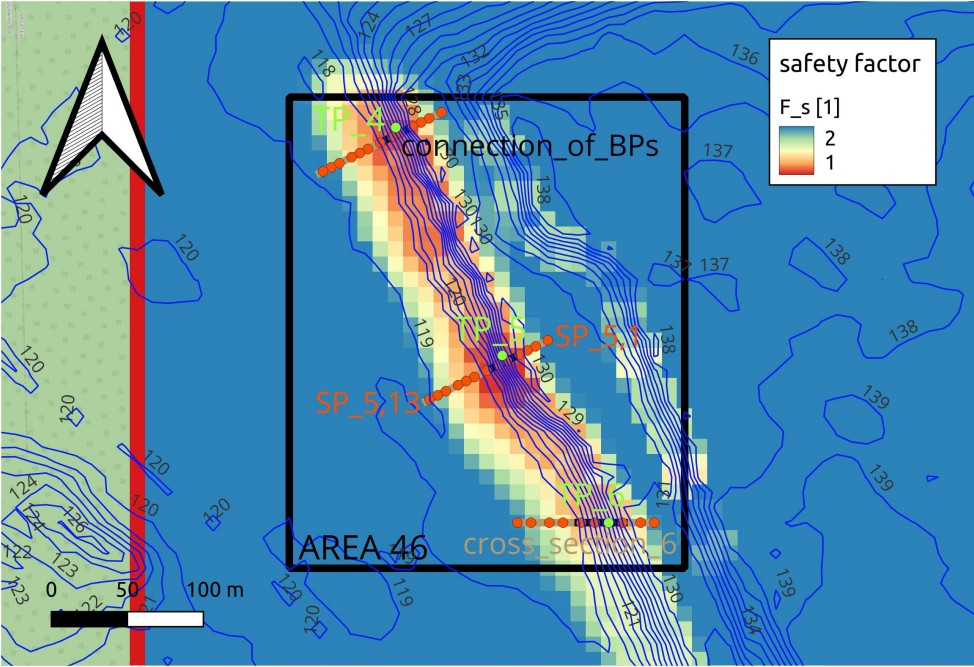

**Figure 18.** Safety factor predicted by KurZeS for non-rooted soil.

For the scenario with non-rooted soil, GEO5 evaluated the safety factor $F_s(TP_5) = 1.08$. Figure 19 shows a 2D cross-section with visualization of the data extracted for complementary safety factor assessment.

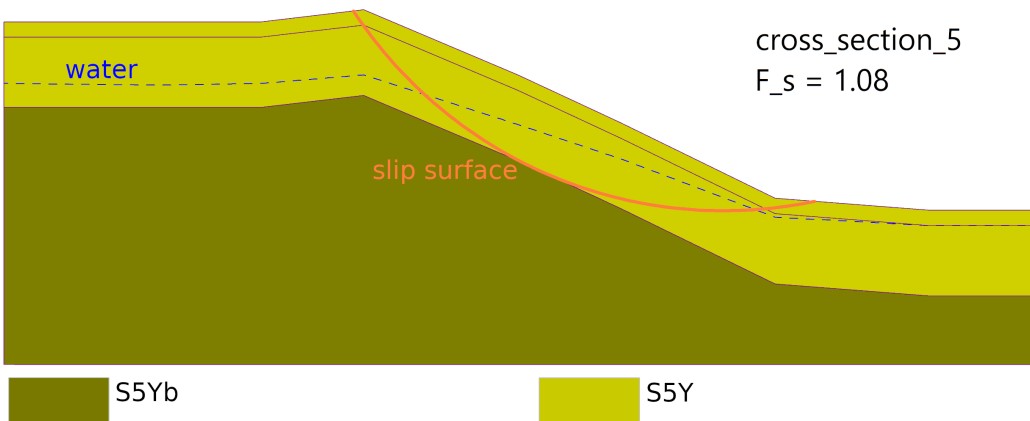

**Figure 19.** Safety factor predicted by GEO5 for non-rooted soil.

For the model with rooted soil, the situation is very similar. KurZeS predicted a safety factor $F_s(TP_5) = 1.18$. Figure 20 depicts the results evaluated for the whole "AREA 46". Safety factor $F_{sR}(TP_5)$ in Figure 20 evinces evident improvement in comparison with Figure 18.

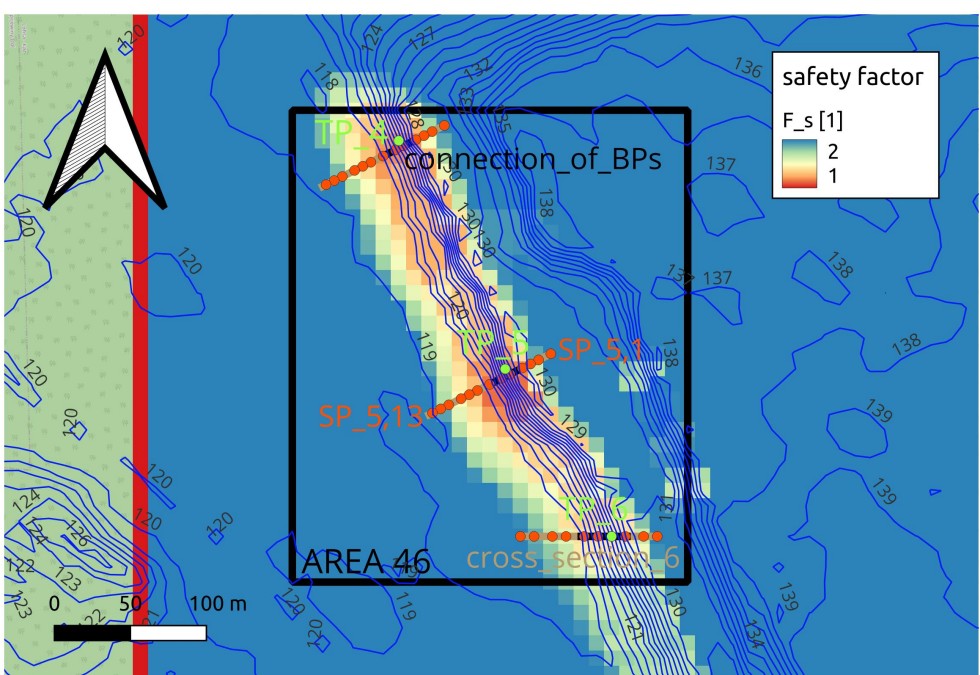

**Figure 20.** Safety factor predicted by KurZeS for rooted soil. TP_i denotes i-th Testing Point and SP_i,j denotes j-th Sampling Point lying on the i-th cross-section.

GEO5 evaluated the safety factor $F_{sR}(TP_5) = 1.16$ for the rooted soils. Visualization of the input data extracted along cross_section_5 is shown in Figure 21.

The difference between the resulting safety factors obtained from KurZeS and GEO5 can be denoted as negligible.

This comparison of two very close results from two different software tools verified the function of KurZeS for the case of a model without surface water.

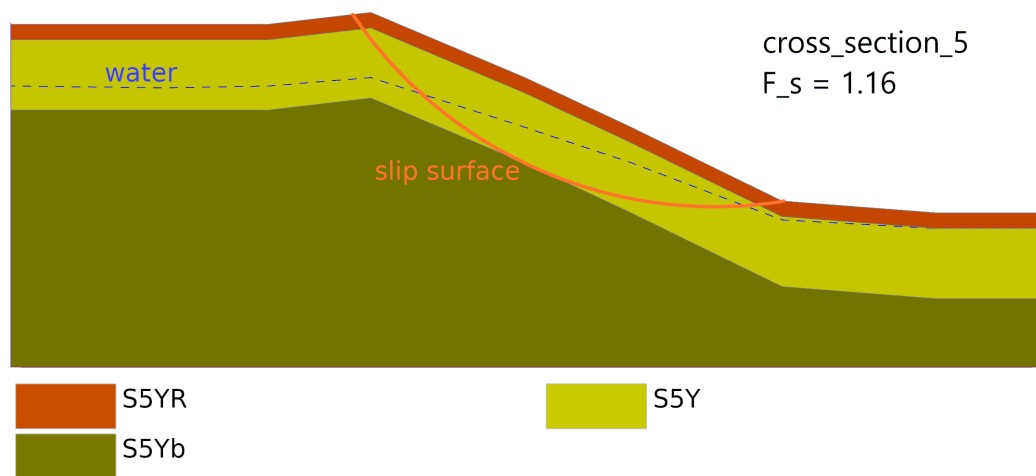

**Figure 21.** Safety factor predicted by GEO5 for rooted soil.

### 3.4.2. Cross-Section 8

Another subarea selected for verification of the results takes place at the northeastern slope of the mining waste dump Scheibe and represents very steep slopes finishing in surface water, see Figure 22. Here, we present the comparison of safety factor predictions at Test Point $TP_8$, which lies directly on cross-section 8.

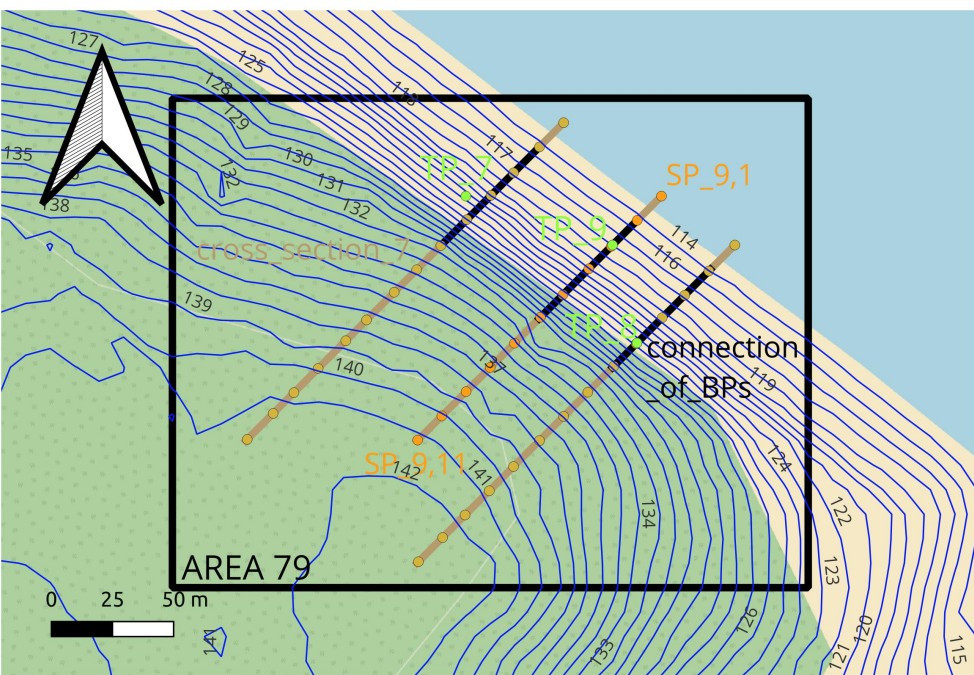

**Figure 22.** Map of AREA 79. TP_i denotes i-th Testing Point and SP_i, j denotes j-th Sampling Point lying on the i-th cross-section.

The KurZeS approach predicted the safety factor $F_s(TP_8) = 1.24$ for slopes covered by non-rooted soil. You can see the results for the whole "AREA 79" in Figure 23. The direction of cross-section 8 is very close to fall lines (almost perpendicular to contour lines).

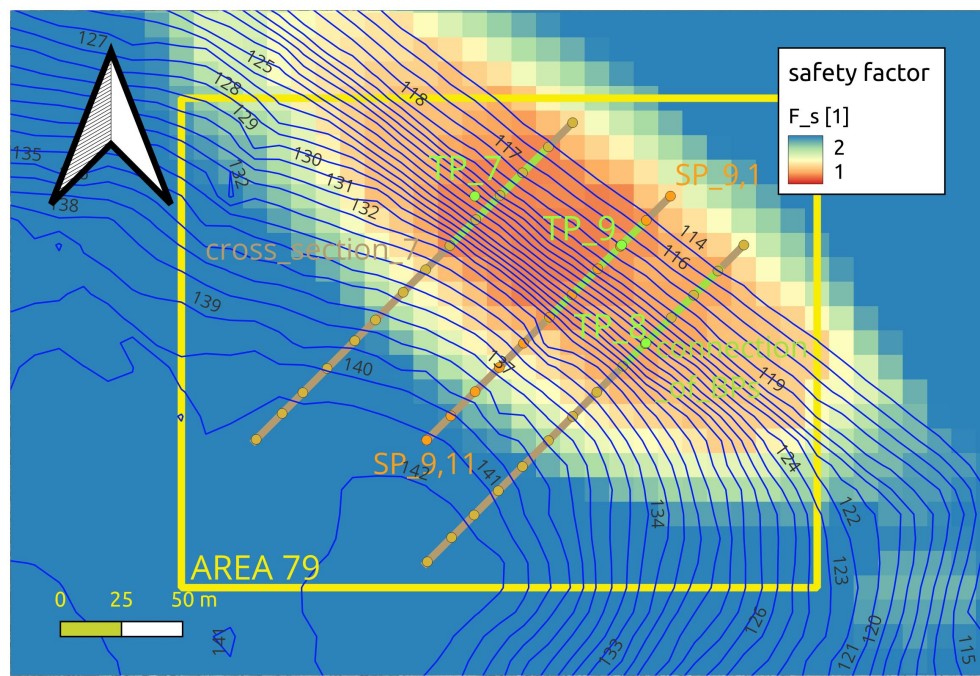

**Figure 23.** Safety factor predicted by KurZeS for non-rooted soil. TP_i denotes i-th Testing Point and SP_i, j denotes j-th Sampling Point lying on the i-th cross-section.

The safety factor of the slope covered by non-rooted soil and evaluated by GEO5 has a value $F_s(TP_8) = 1.25$. In Figure 24, you can see a 2D cross_section_8, which was extracted from input files for KurZeS and evaluated in GEO5.

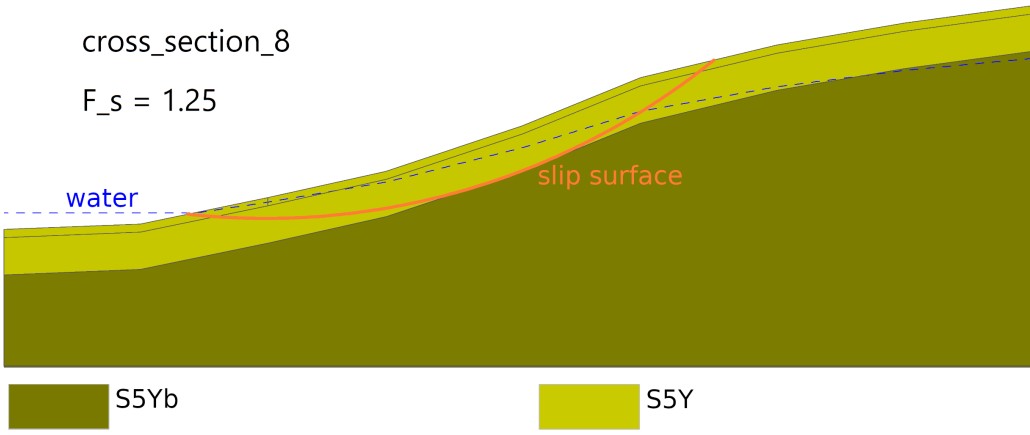

**Figure 24.** Safety factor predicted by GEO5 for non-rooted soil.

The difference between safety factors predicted by KurZeS and GEO5 for slopes formed by non-rooted soils is even smaller than in the case evaluated in "AREA 46" and it can be denoted as negligible.

For the case of a slope formed by rooted soil, KurZeS predicted safety factor $F_{sR}(TP_8) = 1.30$. The resulting raster for the whole "AREA 79" can be seen in Figure 25. Lighter colors than in Figure 23 indicate an improvement in slope stability due to root reinforcement.

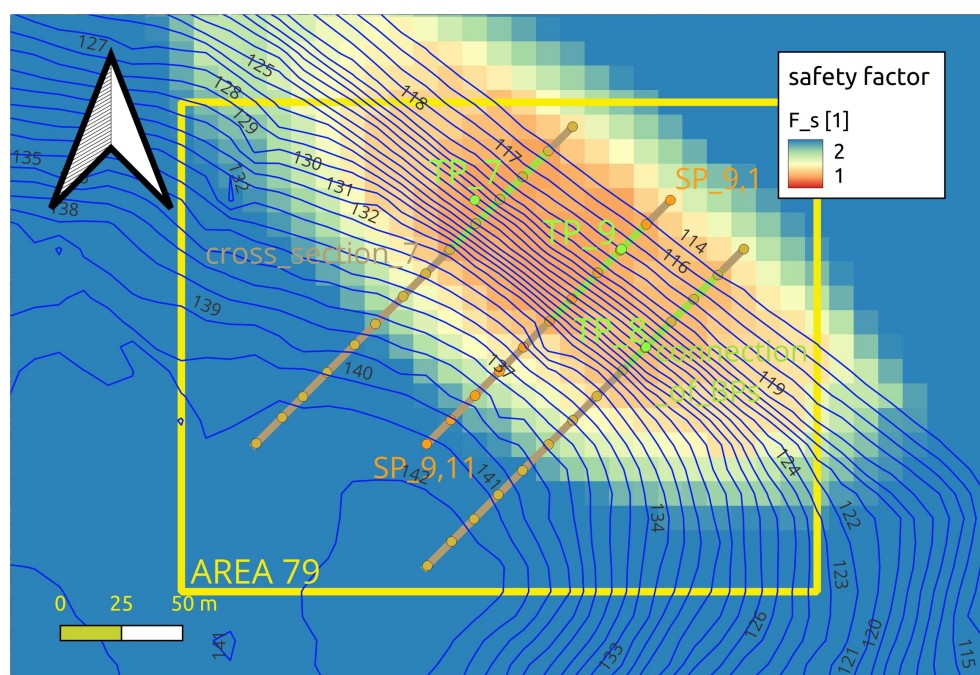

**Figure 25.** Safety factor predicted by KurZeS for rooted soil. TP_i denotes i-th Testing Point and SP_i, j denotes j-th Sampling Point lying on the i-th cross-section.

Verification result obtained for the case of rooted soil from GEO5 has the value $F_s(TP_8) = 1.27$ and the appropriate cross_section_8 with rooted soil is depicted in Figure 26.

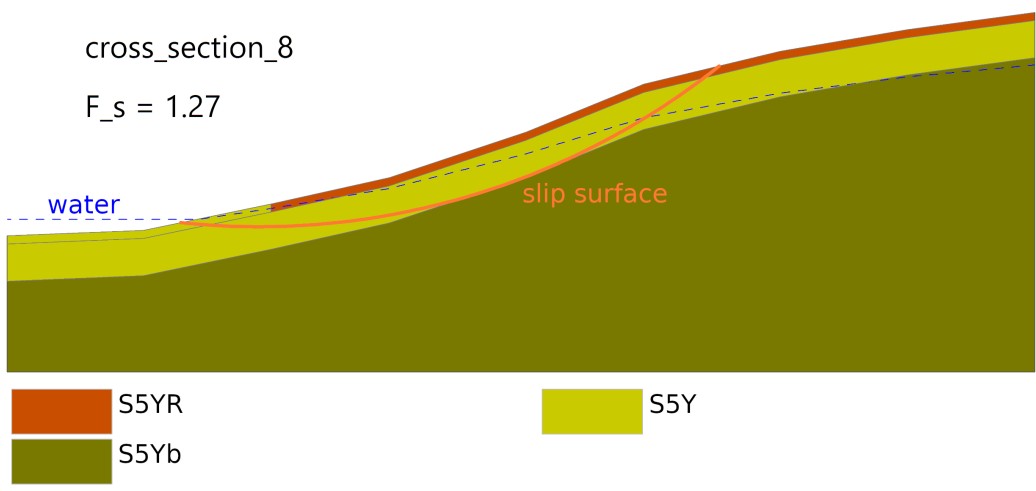

**Figure 26.** Safety factor predicted by GEO5 for rooted soil.

For the case of a slope formed by rooted soil, the difference between safety factors obtained from KurZeS and GEO5 is slightly higher than in previously evaluated cases and it makes for approximately 2.3%, which we feel to be still very low.

This comparison of two very close results from two different software tools verified the function of KurZeS for the case of a model including surface water.

The above-mentioned results show that the results of KurZeS using the pre-calculated database are very comparable with the results of GEO5 for the same slopes. The analysis was conducted both for slopes finishing in surface water and slopes finishing in land, and both for cross-sections including their Test Point and cross-sections that passed it. The comparability of the results verifies the approach using the pre-calculated database as

a good simplification of the problem. This approach efficiently allows many stability factor evaluations for each Test Point and finds the weakest slope in its vicinity.

## 4. Discussion

Commercial software tools for slope stability evaluation have offered the "Method of Slices" to compute safety factors in 2D slope cross-sections. The KurZeS approach uses the "Method of Slices" implemented in GEO5 for pre-computed interpolation tables for selected soil types. These tables are used to approximate the slope stability of large areas.

The use of a pre-computed database allows the KurZeS approach presented in this article to bring an original geometrical and geotechnical concept, where slope stability in each Test Point is not evaluated just along the fall line but also along rather different directions. This concept takes into account more slopes and assigns the TP the lowest safety factor in its vicinity. This "looking around" the TP approach could be important, especially in places where surface water or material interface appear. In such a place, the weakest slope may not be the one that runs exactly along the fall line passing the Test Point.

As in all applications, the accuracy must be balanced with simplicity. The price that should be paid for achievable large-scale models is a simplified two- (case without NBS) or three- (case with NBS) layer stratigraphy, which has been evaluated as sufficient detail for the soil dump slopes under consideration.

In most cases, the simplifications and inaccuracies of the method are related to the mesh discretization density and schematization of the model. In the geometric part, slip surfaces shorter than 20 m are not allowed.

The method allows the $BP_1 - BP_2$ line to pass the TP. The effect of this feature on the accuracy of the result is largely limited by the fact that the least favorable slip surfaces described by both pairs of line segments $BP_1 - TP$ and $TP - BP_2$, or the line segment $BP_1 - BP_2$, usually have the minimum angular deviation from the slope gradient and are close to each other in shape according to the experience analysis. This means that the angle between $BP_1 - TP$ and $TP - BP_2$ is close to $180°$, and therefore, the distance between TP and $BP_1 - BP_2$ is small. Examples of cases with different probabilities of finding the least favorable slip surface with a comparison of the degree of inaccuracy are shown in Figure 27.

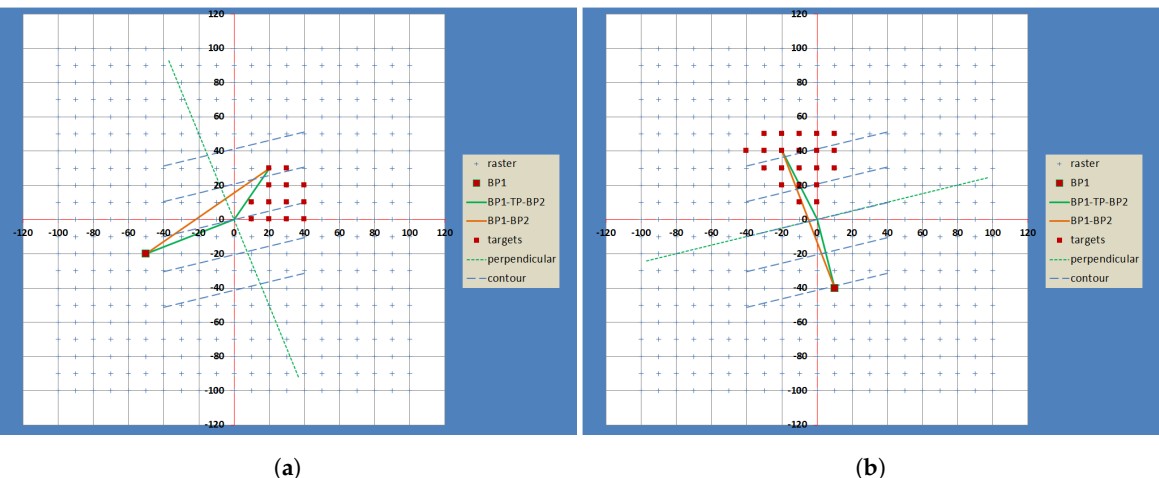

(**a**)                                                    (**b**)

**Figure 27.** (**a**) Low probability that the link would represent the area with the lowest $F_s$; (**b**) a high probability of the same.

The inaccuracies of the method resulting from the influence of the mutual rotation of the $10 \times 10$ m mesh and the gradient direction increase with the decreasing length of the slip surfaces. Therefore they are analyzed in greater detail for the cases of the shortest slip surfaces with a changing angle between the contour lines and the orientation of the $10 \times 10$ m mesh. For this analysis, a completely homogeneous model is considered, without the

influence of groundwater. Therefore, only the slope of the terrain on the line and its length have influence.

For TP $\neq$ BP$_1$ and the 120° angle, there are only two patterns with three lines (Figure 28, the first and second columns of three) with a matching BP$_2$, that are repeated four times. In the case of TP $\equiv$ BP$_1$, there is one pattern repeated four times (third column).

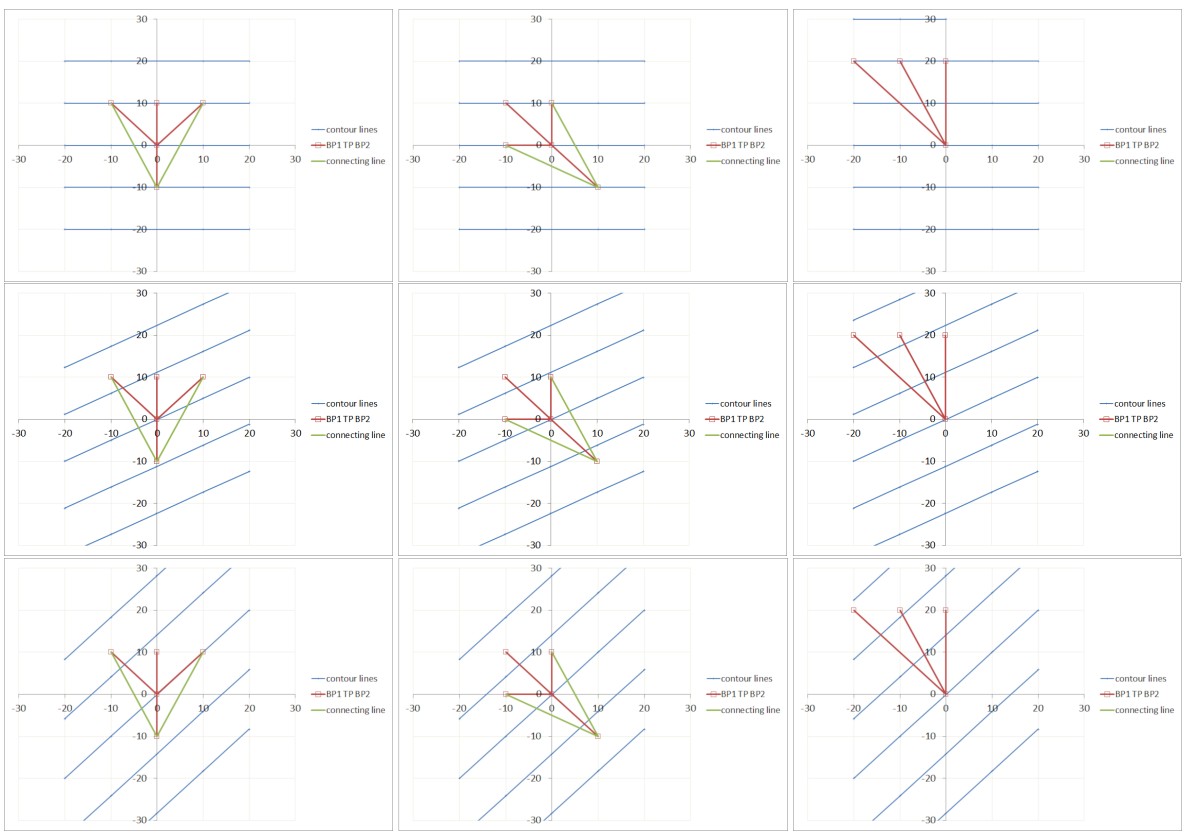

**Figure 28.** Analysis of the influence of the mesh rotation relative to the slope for short links, variants for 120° angle.

In the first triad, the contour lines have the same direction as the horizontal X axis and the slope as the vertical Y axis. In the second triplet, these lines form an angle of 26, 6° and in the third, 45°. One of the possible links for TP $\neq$ BP$_1$ and TP $\equiv$ BP$_1$ is always perpendicular to the contour lines, and its steepness is maximal under the given conditions, but its length changes by leaps from 20 m to 22.36 m ($10 \times \sqrt{5}$) and 28.28 m ($10 \times 2 \times \sqrt{2}$)—Figure 29.

In favor of the adherence of the method to reality, the regularities in terrain modelling, especially in the case of reclamation, the "inverse proportion" of the slope gradient and length are at work. The longest slope cannot be the steepest one.

Further, the potential slope stability improvement achieved through root reinforcement represented by additional root cohesion was studied. Tree species recommended for reclamation of brown coal dumps are birch (*Betula*), larch (*Larix*), red oak (*Quercus rubra*) and pine (*Pinus*) [39]. In the presented simulations, a special type of slope with specific soil composition was considered. This kind of slope is common in soil dumps that are present in brown coal mining areas, e.g., in the Northern Bohemian Brown Coal Basin or Saxony and Lusatia, Germany.

The values of the stability parameter included in the presented model are rather conservative values coming from the literature and an expert's experience. Such a choice of parameters can avoid damage caused by an overestimation of the predicted slope stability.

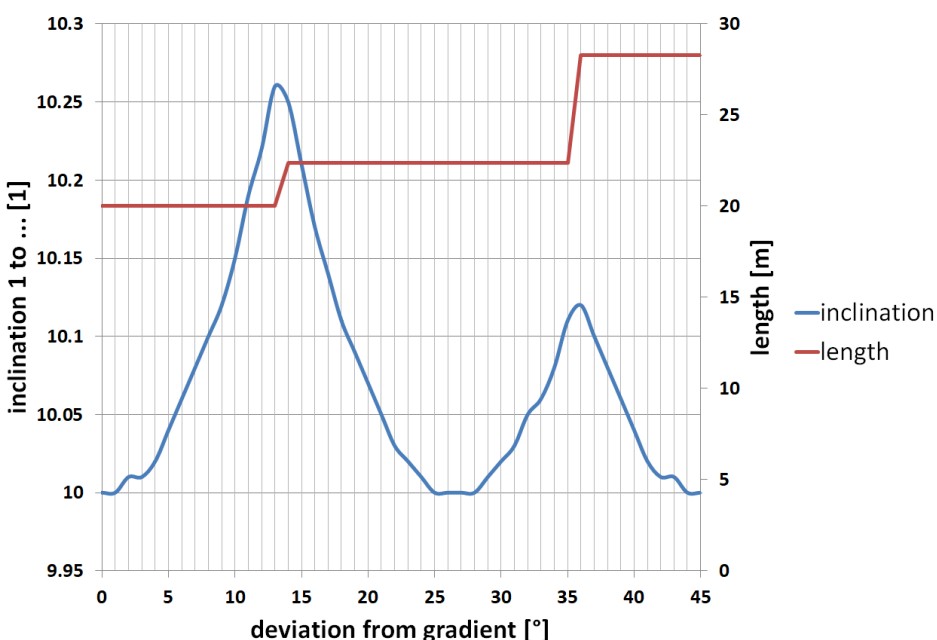

**Figure 29.** Effect of mesh direction rotation relative to fall line for short links.

Predicted slope stability improvements may seem to reach small relative values; however, in very unstable areas where the safety factor approaches the value of 1, the improvement may mean an important difference between the risk and safety.

In this study, neither rainfall [16,18,48,49] nor groundwater flow [50] were considered. These effects could be taken into account by coupling the KurZeS method with different appropriate model tools.

A probable plant-induced water table decrease was not considered. The expected stabilization effect is in the order of a few percent. However, this small improvement may play a crucial role in the boundary cases where the slope is near to collapse.

Temporal dependency of the stabilizing effect, which may reach its optimum after a couple of decades, was also not considered. It could be implemented using time-dependent stability parameters of the soils.

For the sake of simplicity, we have also not considered in the model, the effect of earthquakes, as in [51,52].

In this study, the volume of soil removed by landslides was not predicted. The description of this phenomenon can be found in [8,47,53] and possible methods of its evaluation are described in [54,55].

Before the stabilization effects of the industrial forest fully appear, another NBS (fascines, grass, shrub) may be applied to avoid the risk of shallow landslides and erosion. On a small scale, at specific places, NBS may be complemented with standard technical solutions used for slope stabilization. The same holds for the cases where NBS are not able to stabilize the slope in sufficient measures and where the residual risk of landslides remains too high.

**5. Conclusions**

The presented method allows the geotechnical stability of large areas to be analyzed with generally accessible data on a usual computer in a reasonably short time, which was demonstrated on a specific area of soil dump of the former open-cast mine Lohsa II. It also allows evaluation of the influence of possible natural-based or technological measures, which was demonstrated in the case of root reinforcement.

The two most original aspects of the presented approach are the use of a pre-computed database of safety factors of slopes of various parameters and the testing of the stability of the whole neighborhood of each Test Point (the associated slopes do not need to cross their

TPs). Both these aspects are somehow connected since the pre-computed database shortens computational time so taking more possible slopes into account is allowed.

A compromise between the detail and the large area of the study area is the choice of a square network with a node spacing (TP) of 10 m and a limitation of the shortest slip surface length to 20 m. However, the method includes, to a controlled extent, a rather detailed examination of the immediate vicinity of the TP by also considering links that do not directly pass through the TP. By taking into account slopes, directions, slip surface lengths and other parameters, the method is suitable for rugged terrain with variable soil composition or in contact with water bodies, despite the limitations.

The possibility of the inclusion of NBS was presented in the study of the influence of possible safety factor change due to vegetation (root reinforcement) that can be performed on a computer to help identify the slopes that can be stabilized by root reinforcement and do not need other (specifically technological) measures.

The presented approach was verified and validated in the area of soil dump of the former open-cast mine Lohsa II in Lusatia, Germany. Verification was performed by comparing several slope stability computations obtained from the KurZeS method and GEO5 software. The validation was performed by comparing the KurZeS results with the published map of areas closed to the public.

The maximum predicted relative increase in the stability factor due to the consideration of nature-based solutions was 17% and was observed for extremely short slopes with a length of 20 m or less. Shorter slopes are not accounted for by the model due to its geometrical concept.

However small this improvement may appear, it can be crucial in cases where the stability factor is close to 1.0, i.e., when the driving and motion-resisting forces are close to equilibrium.

The predicted maximum improvement exceeded expectations based on reference calculations from GEO5. This was due to the choice of the minimum tested slope length in the reference calculations being too high.

In further research, the authors will focus on the generalization of quantification of slope stabilization effects achieved through various NBS for various soil compositions. In the application, the focus will be directed towards the simulation of the stability of selected, large-scale mining waste dump slope areas. This will enable us not just to assess risks and threats but also to evaluate the potential to avoid shallow landslides using NBS.

**Author Contributions:** Conceptualization, L.Z.; Methodology, J.K.; Software, L.Z. and J.K.; Validation, J.Š.; Resources, L.Z.; Writing—original draft, L.Z., J.Š. and J.K.; Writing—review & editing, J.Š. and J.K.; Visualization, J.Š.; Supervision, L.Z.; Project administration, L.Z. All authors have read and agreed to the published version of the manuscript.

**Funding:** This research received no external funding.

**Data Availability Statement:** The original contributions presented in the study are included in the article, further inquiries can be directed to the corresponding author.

**Conflicts of Interest:** The authors declare no conflict of interest.

## Abbreviations

| | |
|---|---|
| BP | Boundary point |
| DEM | Digital elevation model |
| $F_s$ | Safety Factor |
| GIS | Geographic Information System |
| LfULG | Landesamt für Umwelt, Landwirtschaft und Geologie |
| MSL | Mean Sea Level |
| NBS | Nature-based solutions |
| SWB | Soil and Water Bioengineering |
| TP | Test Point |

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
