# Peer review of "Inclusion of Nature-Based Solution in the Evaluation of Slope Stability in Large Areas"

_land, doi:10.3390/land13030372_

Round 1

Reviewer 1 Report (Previous Reviewer 1)

Comments and Suggestions for Authors

COMMENTS TO THE AUTHORS:

I read carefully the manuscript entitled: "Inclusion of Nature Based Solution in the Evaluation of Slope Stability in Large Areas". Paper

can be accepted after major revision. The following comments will be beneficial to modify the scientific quality of the manuscript.

-The validation is weak. It is highy recommended to give more emphasis to more validation of this research paper in a subsection

separately.

-How the parameters of the models are determined in the table of the parameters? these specific parameters are not well explained

inTable, what is the rationality behind the range of these variable parameters?

-The introduction is generally. It is desired to rephrase this part to emphasize the objective of the study. In this reviewer's opinion, the

introduction should be rewritten.

-In my view, the numerical modeling has some flaws. Therefore, It is critical to the authors send me main of the numerical modeling file

to modeling be checked. Else I can not accept this resaerch work in next time.

-The technical English is moderate and it should be improved in next submission. It is strongly recommended to re-organize the text and

sections. It is preferred to professionally revise the paper regarding editorial and grammatical rules by particularly a native English

reviewer.

-"Slope failures around the world pose a major challenge to geotechnical engineers as it leads destruction of properties, and sometimes

deaths. Researchers have focused a lot of emphasis on slope failure analysis caused by weather events, earthquakes, groundwater

movement, etc. [1-5]." Many past studies focused on slope stability from the perspective of earhquakes and dynamic analysis. The

authors should include a series of studies focused on investigating slope stability by considering seismic analysis in the introduction

section along with relevant references.

-The novel contributions of this study are not addressed well in the text. It is required to be reorganized in order to become more

apparent and compatible with the manuscript.

-What distinguishes this study from others that have been published in the scholarly literature? As there are numerous other studies in

the body of research that are comparable to these studies, kindly compare them. Write out the similarities and differences, with an

emphasis on your uniqueness. It is crucial to emphasise in the beginning how special this piece of study is.

-The conclusions section should be improved. What should be the benefits/results of this article? The evaluation of the results obtained

can be shortened in the conclusions section. Also, author should highlight any limitations of your study, describe future directions for

research and recommendations in conclusions section.The conclusions section is suggested to be written more concisely, especially,

excluding those that have been well known to researchers and designers.

-The references list is composed of several writing styles and punctuation errors, which is unacceptable. For instance, some are not,

some have DOI, and others do not and etc.

Upon the above-mentioned comments, I can recommend publishing the article only after significant improvements.

Comments on the Quality of English Language

The quality of English language is moderate.

Author Response

Reviewer 2 Report (Previous Reviewer 2)

Comments and Suggestions for Authors

This study has strong practical application value, the article is well structured and the methodology is mature.

Suggestions for further improvement and modification:

1. The introduction part can be further expanded, and it is recommended to further emphasize why this research is done, what new problems need to be solved in this research, and what are the innovations.

2. In Figure 1, the legend cannot cover the red range line.

3. In the methodology section, the length can be streamlined to reduce the process document presentation of the case study.

4. In the results section, the graphs can be appropriately grouped.

5. In the discussion part, it is recommended that based on the analysis results, an in-depth discussion of specific measures to enhance slope stability.

6. Move the discussion related to the shortcomings of this study into the conclusion part.

Round 2

Reviewer 1 Report (Previous Reviewer 1)

Comments and Suggestions for Authors

Comments on the Quality of English Language

The quality of the English language is moderate.

Reviewer 2 Report (Previous Reviewer 2)

Comments and Suggestions for Authors

The quality of this article has been improved to a certain extent.

This manuscript is a resubmission of an earlier submission. The following is a list of the peer review reports and author responses from that submission.

Round 1

Reviewer 1 Report

Comments and Suggestions for Authors

I read carefully the manuscript entitled: "COMPUTATIONAL ASSESSMENT OF SLOPE FAILURE RISKS."
The following comments will be beneficial to modify the scientific quality of the manuscript.
-This manuscript has been submitted as a original Paper but I think this is not the appropriate article
type. In my opinion the authors should revise this research as aTechnical Note or add more parametric
studies in paper else, in my view this research should reject in next round. Now, this research work is
simply a report.
-It is critical to validate the results against observations. Through this way, one could know the validity
of the models. The authors should add a validation section along with related diagrams and with fully
technical explanations. A final comparison of the your laboratory results with numerical modeling or...
will reveal that results of the paper are correct. This matter is very important me. Include a separate
section titled verification and then verify results using related diagram.
-Article title is not suitable. It should be changed.
-Rewrite the abstract. The abstract should explain the aims and scope of the work presented in the
article rather than introducing the research topic, thus, the abstract should be condensed. Keep the
abstract short and concise. The first sentence should motivate your study, then explain clearly and
concisely what you did and at the end your main result and conclusion and its importance/impact.
-I think that the manuscript is not very innovative overall. The novel contributions of this study are not
addressed well in the manuscript. It is required to be reorganized in order to become more apparent.
-There are some gramatical errors in the writing.The authors should check the spelling and grammar
of the manuscript.The writing style of the paper poor and should be improved. Else, I will reject this
paper in next round.
-The introduction section should be rewritten. Now, The introduction is generally. It is desired to
rephrase this part to emphasize the objective of the study. The introduction should be rewritten and
extended with more and new refs.
-The structure of the paper should explain in the last paragraph of the introduction.
-Although the paper objectives are mentioned in introduction part, additional information are provided,
making it difficult to comprehend these objectives. It should be uniquely stated what the aim is then
the importance should be appreciated by those who are interested in this paper. I suggest splitting
this part into two paragraphs or numbering the objectives, in order to be clear. Please, apply.
-This literature survey of the present study is very much limited, please define new contribution of the
present study which was not covered in the past literature.
-Many past studies focused on slope stability from the perspective of wall movements using
numerical modeling or laboratory methods. The authors should include a series of studies focused on
investigating slope stabiluty in the introduction along with relevant references. 

-What distinguishes this study from others that have been published in the scholarly literature? As
there are numerous other studies in the body of research that are comparable to these studies, kindly
compare them. Write out the similarities and differences, with an emphasis on your uniqueness. It is
crucial to emphasise in the beginning how special this piece of study is.
-The authors should improve discussion sections. Clearer presentation and discussion of the results
should be provided. The reported claims should be adequately discussed in the context of the
literature.
-How the parameters of the Tables have been determined? these specific parameters are not well
explained inTables, what is the rationality behind the range of these variable parameters?
-Results and discussion section has been considered very brief compared to other sections of the
article.The authors should include more parametric studies in this research work.
-All of the parmeters in text, figures, and tables, ... should be italics. Check all of the parameters.
-Some text within the figures is too small, please revise them based on the requirement of the journal.
-The quality of the sone figures is poor. Modify all of them.
-All of the equations and parameters should be checked again.
-The conclusions section is suggested to be written. specially, excluding those that have been well
known to researchers and designers.
-In conclusion section authors should highlight any limitations of your study, describe
recommendations.
-The references, and how they are cited in the text, should be checked again.
- References are incomplete and written in an arbitrary fashion composed of several writing styles and
punctuation errors. Moreover, these are small in number and can be increased.
Upon the above-mentioned comments, I can recommend publishing the article only after above
minor improvements

Comments on the Quality of English Language

The quality of English language is moderate and can be improved.

Reviewer 2 Report

Comments and Suggestions for Authors

This article examines computer-aided slope stability assessment. Overall, this research is of interest. The following are suggestions for revision and improvement:

(1) Increase the sources of the literature review, especially the research base in this field in other parts of the world.

(2) There are only a few subsections 2.1 in Part 2, and it is suggested to reorganize the description of the content of this part.

(3) Regarding the Materials and Methods section of Part III, there are no particular comments in general.

(4) Regarding the results of part 4, the problems of Part 2 still exist, and it is recommended to reorganize the content structure to present the results of the analysis more clearly.

(5) Regarding the discussion in Part 5, the depth and organization of the existing content is insufficient and optimization is recommended.